# Potential Impacts of Certain N_2_-Fixing Bacterial Strains and Mineral N Doses for Enhancing the Growth and Productivity of Maize Plants

**DOI:** 10.3390/plants12223830

**Published:** 2023-11-11

**Authors:** Moustafa Shalaby, Mohssen Elbagory, Nagwa EL-Khateeb, Ahlam Mehesen, Omaima EL-Sheshtawy, Tamer Elsakhawy, Alaa El-Dein Omara

**Affiliations:** 1Agricultural Botany Department, (Agricultural Microbiology), Faculty of Agriculture, Kafrelsheikh University, Kafr El-Sheikh 33516, Egypt; meshalaby66@gmail.com (M.S.); nagwa.elkhateeb@yahoo.com (N.E.-K.); omimaelsheshtawy@gmail.com (O.E.-S.); 2Department of Biology, Faculty of Science and Arts, King Khalid University, Mohail 61321, Assir, Saudi Arabia; mhmohammad@kku.edu.sa; 3Agriculture Microbiology Department, Soils, Water and Environment Research Institute, Agricultural Research Center, Giza 12112, Egypt; ahlam.mehesen@gmail.com (A.M.); drelsakhawy@arc.sci.eg (T.E.)

**Keywords:** *Azotobacter*, *Azospirillum*, 16S rRNA, enzyme activities, maize production

## Abstract

The enhancing effect of N_2_-fixing bacterial strains in the presence of mineral N doses on maize plants in pots and field trials was investigated. The OT-H1 of 10 isolates maintained the total nitrogen, nitrogenase activities, IAA production, and detection of NH_3_ in their cultures. In addition, they highly promoted the germination of maize grains in plastic bags compared to the remainder. Therefore, OT-H1 was subjected for identification and selected for further tests. Based on their morphological, cultural, and biochemical traits, they belonged to the genera *Azotobacter*. The genomic sequences of 16S rRNA were, thus, used to confirm the identification of the genera. Accordingly, the indexes of tree and similarity for the related bacterial species indicated that genera were exactly closely linked to *Azotoacter salinestris* strain OR512393. In pot (35 days) and field (120 days) trials, the efficiencies of both *A. salinestris* and *Azospirillum oryzea* SWERI 111 (sole/dual) with 100, 75, 50, and 25% mineral N doses were evaluated with completely randomized experimental design and three repetitions. Results indicated that N_2_-fixing bacteria in the presence of mineral N treatment showed pronounced effects compared to controls. A high value of maize plants was also noticed through increasing the concentration of mineral N and peaked at a dose of 100%. Differences among N_2_-fixing bacteria were insignificant and were observed for *A. oryzea* with different mineral N doses. Thus, the utilization of *A. oryzea* and *A. salinestris* in their dual mix in the presence of 75 followed by 50% mineral N was found to be the superior treatments, causing the enhancement of vegetative growth and grain yield parameters of maize plants. Additionally, proline and the enzyme activities of both polyphenol oxidase (PPO) and peroxidase (PO) of maize leaves were induced, and high protein contents of maize grains were accumulated due to the superior treatments. The utilization of such N_2_-fixing bacteria was, therefore, found to be effective at improving soil fertility and to be an environmentally safe strategy instead, or at least with low doses, of chemical fertilizers.

## 1. Introduction

Maize (*Zea mays* L.) is considered a highly prized plant due to its economic significance, alongside its nutritional value and sensory attributes. It is the main source of human foods, animal feed, and raw materials in manufacturing [1]. The productivity of maize grains in Egypt reached 6.4 million metric tons during 2021 in comparison with 6.8 MMT during the 2018 season. Egypt is the world’s 14th biggest maize producer, and its harvest comes from two summer planting cycles [2]. Maize plants need high rates of mineral fertilizer inputs, notably fertilizer-N, which is the most effective way to increase agricultural output [3]. In contrast, the overuse of such fertilizers causes pollution for soil, water, and air, in addition to increasing the cultivation cost [4].

The potential for employing natural and safe compounds to counteract the negative effects of chemical fertilizers was recently spotlighted. These natural components could also promote the plants’ growth and are called “plant-friendly microorganisms”, which live in or near plants and can promote development through a variety of processes, including increased production of phytohormones, defense against biotic and abiotic stressors, and enhanced nutrient and water intake [5]. Based on their compatibility and combined efficacy, both in vitro and in vivo, the most effective strains of beneficial microorganisms for the planting situation must be identified. This consortium of microorganisms must then be used in actual agricultural situations for efficient management and production to promote plant growth and soil health via seed treatment or foliar application [6,7].

The combined inoculation of (two or more) beneficial and compatible organisms performs better than single inoculants, suggesting that combined inoculations could more effectively interact with the introduced organisms [8,9]. Combination inoculations are thought to be more effective than the single one. It might have a greater inhabitant diversity and population, which will result in higher levels of interaction and thereby a more stable environment and better nutrient transformation [10]. In addition to increasing plant growth when single inoculation was employed, co-inoculation enhanced plant growth and physiology, including photosynthesis, the amount of chlorophyll in leaves, and enhanced plant uptake of macro- and micronutrients like Mo, Fe, and Mg [7,10].

*Azospirillum* and *Azotobacter* species exhibited excellent adaptability bacterial genera to colonize diverse host plants. They have been isolated worldwide and used as bio-fertilizers in the cultivation of several crops [11]. Such microbes were well known as plant growth-promoting rhizobacteria (PGPR) [12]. The biological nitrogen fixation of maize, rice and wheat might be increased by 25–50% when treated with *Azospirillum* and/or *Azotobacter* spp. [13]. Several studies showed that application of bio-fertilizers in combination with chemical ones improved growth and yield of the crops [14,15,16,17].

Therefore, we aimed at isolating, screening, and selecting an effective and promising *Azotobacter* for its favorable features and impacts on plant growth. The effects of simultaneous *Azotobacter* and *A. oryzae* inoculation on the growth and productivity characteristics of maize plants were also investigated in a pot and field experiment, with varying concentrations of mineral fertilizers used.

## 2. Results

From different maize fields representing 7 regions within Kafr El-Sheikh Governorate, 10 N_2_-fixing bacteria were isolated. From the 10 isolates, visualization of the cultures growing on Ashby mannitol medium showed a round brown halo of bubbles of nitrogen gas, indicating a strong probability of being *Azotobacter*.

### 2.1. Optimization of the N2-Fixing Isolates

The 10 isolates were subjected to test their total nitrogen contents, nitrogenase activities, IAA-production, and NH_3_-detection in their cultures. A remarkable pronounciation of OT-H1 (Elhamoul) followed by OS-M1 (Motobas) isolates in these features was noted compared to others (Table 1). Meanwhile, some isolates were characterized in some traits. Among these, OT-B1 (from Balim) had a total N content that reached 10.50%. OT-D2 (Disouq) was ranked at the third position after the superior isolates for nitrogen activity and IAA production.

### 2.2. Germination (%)

The germination percentage of inoculated maize grains with 10 isolates was evaluated in small plastic bags. All 10 tested isolates were found to be a plant growth promoting rhizobacteria (PGPR) by remarkably enhancing the maize grains’ germination compared with controls (Figure 1). OT-H1 followed by OS-M1 were the superior isolates for inducing germination, reaching 96.66 and 93.33% of maize grains, respectively, compared to 73.33% for the control. It indicated that both isolates have a great ability to promote the emergence and growth of maize seedlings (Figure 1). Accordingly, the OT-H1 isolate was subjected for identification and selected for in vivo trials.

### 2.3. Identification of the N_2_-Fixing Isolate

#### 2.3.1. Cultural and Biochemical Tests

Data describing the cultural and biochemical characteristics of N_2_-fixing bacterial isolate are tabulated in Table 2. It shows that the isolate was found to be motile, Gram-negative, and non-endospore-forming bacteria. OT-H1 seemed to exhibit a short rod or oval cell shape, and its colonies were initially white on Ashby’s mannitol agar medium but turned brownish, watery, and mucilaginous after 48–72 h. The isolate was also found to have similar abilities to induce catalase, and phosphate solubilization, besides using glucose, fructose, and mannitol as a sole source of carbon. OT-H1 was found to be a tolerant until pH 9.5, 40 °C, and 4% NaCl were reached. Therefore, OT-H1 isolate is related and closely belonged to genera *Azotobacter* (OT-H1).

#### 2.3.2. Molecular Test

To confirm its identification, genomic sequences of 16S rRNA of *Azotobacter* coded as OT-H1 were tested. Accordingly, the tree and similarity index of the related bacterial species (*Azotoacter*) are plotted in Figure 2. It could be clearly observed that the Azotoacter isolate OT-H1 was exactly linked to the species salinestris, with highest sequence similarities reaching 99.65% for the *A. salinestris* strain. Sequence data were also submitted to GenBank, and it provided a GenBank accession number, OR512393 (Figure 2).

### 2.4. Pots Trial

In pots, the efficiencies of both *A. salinestris* and *A. oryzea* with and without mineral N 35 days from planting were evaluated. The tested treatments played a function in the dry matter of leaves and shoots, increased the area and total nitrogen of leaves, and amended the content of maize seedlings’ chlorophylls a-b (Table 3 and Figure 3). Results indicated that the pronounced effects of the treatments included N_2_-fixing bacteria in the presence of mineral N compared to the control. The enhanced properties of maize seedlings were noticed during the gradual increase in mineral N doses and peaked at 100%, directly sowing a dose-dependent effect. Thus, the dual mixture containing A. salinestris and A. oryzea with 75% N was the superior treatment, after which the maximal amounts of accumulated dried matter reached 95.61 and 92.27% for shoots and leaves, respectively. It might be caused by the largest leaf area (179.70 cm^2^), the highest densities of chlorophylls a and b (37.82 and 20.66 µg/cm^2^, respectively), and the great accumulated value of total-N (1.07%) which happened due to the superior treatment. The differences among N_2_-fixing bacteria were also insignificant, but tiny effects due to A. oryzea compared to A. salinestris in their singular treatments with the different mineral N were noted. On the other hand, the minimal value resulted in the control, indicating a lower metabolic activity (Table 3 and Figure 3).

### 2.5. Field Trials

#### 2.5.1. Growth Parameters

Under open field conditions, the efficiencies of both *A. salinestris* and *A. oryzea* with and without mineral N 65 d after planting were confirmed. The tested treatments played a function in the plant height, leaf dry matter, leaf area, and total nitrogen of flag leaf of maize plants (Table 4). Data obtained in pots were well confirmed under field conditions. The pronounced values of all treatments compared to the control (100% N) one were calculated. For treatments that included mineral N alone, utilization of 100% followed by 75% doses showed the highest values compared to the other concentrations. For the other treatments, N_2_-fixing bacteria were applied instead of 100% mineral N doses. The highest values by 75% mineral N in presence of *A. salinestris* were then achieved, with a lower value than their corresponding treatments in the presence of *A. oryzea* for all parameters, indicating high N_2_-fixing activities. Accordingly, the N_2_-fixing mixture of both bacteria and 75% mineral N was the superior treatment, where the maximal values were noted. Due to the superior treatment, the largest values of 146.70 cm, 85.69%, and 1.35% of plant height, dried matter, and total-N of leaf were gained, respectively. A remarkable enlargement in leaf area (909.72 cm^2^), nearly double that of each bacterium, was reached through applying the superior treatment, referring to the additive effect of both bacteria.

#### 2.5.2. Biochemical Measurements

Sixty-five days after sowing, the biochemical measurements (chlorophyll a-b and proline content) of maize leaves showed significant variations (*p* < 0.05) across different mineral doses of N with and without bacterial inoculation treatments with *A. salinestris* and *A. oryzea* (Figure 4). Under T11 treatment (75% N + Mix), the highest values for chlorophyll a-b and proline content were 56.55, 36.44 µg/cm^2^, and 14.59 µmol g^−1^, higher than those in the control and the other treatment, respectively (Figure 4).

#### 2.5.3. Enzymes Activities

To evaluate their metabolic responsibility, enzyme activities of proline, PPO, and PO were tested after 65 days of planting under field conditions. As illustrated Figure 5, a remarkable induction of the tested enzyme due to all treatments compared to the control was noted.

The superiority of the treatment that included 75% mineral N with a N_2_-fixing dual bacterial mixture for enhancing activities of PPO and PO was confirmed compared to the other treatments. Due to the superior treatment, the maximal values of 1.17 and 5.64 Unit g^−1^ for PPO and PO were induced, respectively. On the other hand, the control treatment (100% N) showed the lowest induction of the tested enzyme activities, indicating less metabolic action. The enzyme activities of the other treatments were located between the two poles of control and the superior treatments (Figure 5).

#### 2.5.4. Yield Parameters

Under field conditions, the effects of the tested treatments 95 days after planting were evaluated. Table 5 shows the progressive values of grain yield parameters in accordance with the nitrogen concentration and high dose with valued productivity. The superiority of the dual culture with 75% mineral N for enhancing maize yield parameters was also confirmed compared to others. A massive number reached 694.00 grains per cob, which leads to 754.33 g in weight of 1000 kernels in the case of the superior treatment. The high nitrogen content and grain protein were, therefore, reached at 7.73 and 10.14% in the superior treatment compared to 1.21 and 0.17% in the control one, respectively.

## 3. Discussion

The complex system known as the rhizosphere is composed of the soil close to the roots, the root surface with its surrounding slime layer, and the endorhizosphere. The microbiological activity of the rhizosphere is influenced by root activities, particularly the exudation of organic substrates like sugars, amino acids, enzymes, vitamins, carbohydrates, and several other substances. These chemical elements and microbial interactions that aid in nutrient availability may have a significant impact on the soil microflora, which in turn may have a significant impact on plant growth and development [18]. Therefore, through modifying the helpful rhizosphere bacteria, crop productivity can be increased. Therefore, an effort was made herein to isolate, screen, and select an effective and promising *Azotobacter* for its favorable features and impacts on plant growth. Meanwhile, the effects of combined *A. oryzea* inoculation on the growth and productivity parameters of maize plants were also fully studied.

For the production of plant growth promoting substances, the most N_2_-fixing bacteria have a great ability to produce IAA and detect NH_3_ in their cultures with excessive amounts of total nitrogen and nitrogenase activity, such as *Azotobacter* species [19,20]. These results agreed with related findings by the authors of [21], who isolated and identified N_2_-fixing bacteria which could grow in higher levels of pH to use them as successful inoculants. The authors of [22] isolated alkalophilic species of *Azotobacter* from the Ratnagiri Sea shore and other locations in India. Enriched samples were prepared from collected saline soil and inoculated on modified Ashbey’s nitrogen free mannitol broth.

For the role of N_2_-fixing bacteria to enhance the germination (%), our results are in agreement with the findings of the authors of [23], who found that most free-living microbes in soil can fix nitrogen, especially during the early growth phase. On the other hand, the remaining isolates caused moderate effects between the two poles, where their germination ranged from 70.00 to 86.66%. Plant growth hormones, the antioxidant system, the production of siderophores, and an improvement in plant nutrition are all often improved with PGPR [24]. *Azotobacter* can help plants to minimize the negative effects of abiotic stresses, and they can survive in the absence of their host due to the presence of poly-βhydroxybutyrate (PHB) and polysaccharide synthesis [25].

On the other hand, the result for identification matched well with the authors of [20], who identified some N_2_-fixing isolates from different soils as *Azotoacter* depending on their cultural and biochemical properties. These results agreed with the authors of [26,27], who stated that the *Azotobacter* isolate was found to be Na^+^-dependent, N_2_-fixing, and plant growth promoter strains. On the other hand, [28] found that some of the isolated bacteria showed remarkable plant growth-promoting traits, including high salt tolerance, N_2_ fixation, P solubilization, and IAA production.

Based on both pots and field trials, superiority of the treatment included 75% mineral N in the presence of both N_2_-fixing bacterial strains might be attributed to enhancing the cell elongation or cell division, indicating plant growth promotion [29]. It was explained by the authors [30], who found that N-stimulating amino acids are required to form the protein of the cytoplasm and activate the enzyme system required for cell division and elongation. The authors of [29] also confirmed that the mixture of biofertilizers, either alone or in combination, can improve the total nitrogen and chlorophyll pigments in maize with only a half dose of the recommended amount of chemical fertilizers. The efficiency of the superior treatment was also explained via “microbe’s capacity”, which describes microbial growth regulators and may be beneficial for plant development through boosting photosynthesis, nutrient translocation, and nutrient accumulation [31]. PGPR were found to play a key role in enhancing stomata conductance and chlorophyll content index in maize seedlings. Through fixing nitrogen, diazotrophs may have higher leaf N levels. The chemical fertilizers are hardly replaced by diastrophic bacteria. Sustainable agriculture may need diastrophic inoculants. Ref. [32] stated that the dual combination of N_2_-fixing bacteria, including *A. oryzae* NBT506 and *Bacillus velezensis* UTB96, caused an enhancement in biomass production and enzyme activities. It indicates that this bacterial mixture could be more suitable and effective for field applications as PGPR in the form of a dual mixture compared to a singular treatment [21]. Through mechanisms associated with enhancing plant development, increasing mineral intake, increasing dry matter, improving water absorption, and increasing yield, *Azospirillum* is advantageous to plants.

The high level of proline in maize plants might be explained as self-defense to adapt with the environmental changes. Ref. [33] found that the increasing enzyme activities and proline content in the presence of PGPR aided plants to keep water homeostasis via maintaining turgor pressure and resisting osmotic stress. Auxin-producing PGPR enhanced the growth of the root system, thereby helping the plants to absorb more water and nutrients and then increasing the high N_2_ fixation and photosynthetic energy levels [34].

For oxidative enzymes, data indicated lower activity levels of PPO compared with PO. Our findings are well matched with those of the authors of [35], who found a pronounced value for PO compared to their corresponding PPO in onion plants. These responses were termed as induced systemic resistance (ISR) [36]. The PO catalyst is one of the most important oxidative enzymes, generating reactive oxygen species (ROS), polymerizing cell wall composites, and regulating H_2_O_2_ levels [37]. Thus, building a rigid cell wall or producing ROS might be possible due to these multifunctional enzymes. It makes a cell wall more malleable so that it can prevent microbial attacks through raising physical barriers or through counterattacking with ROS production. PPO catalysis plays an important role in lignin biosynthesis and other oxidative phenols. Results also agreed with the findings of the authors of [38], who stated that inoculation with *A. lipoferum* or *A. chroococcum* significantly reduced the stress caused by unsuitable conditions of maize plants through improving vegetative traits, the ROS scavenging system, the ion exchange of Na^+^ and K^+^, and then the activity of the redox enzymes. Thus, the significant effects of both *Azotobacter* and *Azospirillium* in the presence of mineral N on dry matter, chlorophyll, enzyme activity, and total nitrogen enhanced all biological functions in maize plants [39]. Here, *A. salinestris* and *A. oryzea,* as PGPR, played a key role, which was confirmed through ISR via inducing several plant defense mechanisms.

For the yield parameters, our data were in line with the findings of the authors of [14,40], who reported that the utilization of both chemical and biological fertilizers increased the amount of maize grains, cob weight, row number, biological yield, grain number/row, cob length, number of rows/cob, and weight of 1000 kernels. In cereals, a negligible yield increase was noted when *A. salinestris* or commercial bio-fertilizers were combined with 50% N dosage. The key role achieved by *Azospirillum*, as PGPR, was also confirmed by the authors of [21], who reported that it is very active and popular nitrogen fixer in laboratories and under field conditions, causing improvements in the growth and yield of different agricultural crops. The authors of [41] revealed that using growth-stimulating bacteria, the use of chemical fertilizer may be decreased by 50% without compromising maize production.

## 4. Materials and Methods

### 4.1. Soil Sampling

To isolate N_2_-fixing bacteria, soil samples were collected from rhizosphere regions of maize cultivations of Baltim, Desouq, Elhamoul, Motobas, Qulin, Sakha, and Side Salem located in Kafr El-Sheikh Governorate, Egypt, as described in Table 6. Samples were wrapped in polyethylene bags in an ice box and brought to the laboratory at the Soils, Water and Environment Research Institute, Sakha Agricultural Research Station, Kafr El-Sheikh, Egypt. The pH and EC of the 7 samples were estimated based on the methods described in [42].

### 4.2. Isolation of N_2_-Fixing Bacteria

A total of 10 g of each soil sample were diluted to complete 100 mL in 250 mL Erlenmeyer flasks using sterile dH_2_O. The soil solution was shaken at 150 rpm at 30 ± 2 °C for 30 min. For *Azotobacter*, 1.0 mL of the soil solution was transferred to successive dilution tubes containing sterilized broth medium of Ashby mannitol and incubated at 30 ± 2 °C for 15 d. The appearance of brown halo bubbles on the liquid medium indicated the presence of nitrogen gas. Spreading 50 µL of each dilution twice over Ashby mannitol agar medium was performed using the dilution–pour plate technique (10^−1^ to 10^−5^). Media consisted of 20.0 g mannitol, 0.2 g K_2_HPO_4_, 0.2 g MgSO_4_·7H_2_O, 0.2 g NaCl, 0.1 g K_2_SO_4_, 5.0 g CaCO_3_, and 15.0 g agar per 1000 mL, and we adjusted their pH at 7.0 ± 2.0 [43]. Loops of the cultured broth media were streaked on the surface of agar plates and incubated at 30 ± 2 °C for 5 d. The formation of sticky brown colonies on the agar surface indicated the presence of *Azotobacter*. Purified isolates were enriched through streaking them on respective slants containing 50% (*w*/*v*) glycerol and kept at −20 °C for further use [44].

### 4.3. Optimization of the N2-Fixing Isolates

Ten N_2_-fixing bacterial isolates were resulted and subjected to test their efficiencies. Total nitrogen contents, nitrogenase activity, the production of indole acetic acid (IAA), and the detection of NH_3_ were determined in the bacterial cultures.

#### 4.3.1. Total Nitrogen Content

Total nitrogen was measured in 10.0 mL of Jensen’s liquid medium inoculated from 1.0 mL (1.5 × 10^8^ CFU/mL) of the inoculum of freshly growth cultures on a rotary shaker (150 rpm) at 30 ± 2 °C for 48 h. Using the micro-Kjeldahl methods, total nitrogen was determined based on the formula applied in [45] as follows:Total nitrogen (%) = ((V − V0) ∗ N ∗ 0.014007 ∗ 100)/W)(1)
where V0 = blank titration volume (mL), V = sample titration volume (mL), N = normality of standard acid (H_2_SO_4_), and W = sample weight (g).

#### 4.3.2. Nitrogenase Activity

Bacterial isolates were grown in 20.0 mL of Jensen’s liquid medium in little bottles with 2.0 mL of the inoculum of freshly growth cultures from each isolate at 30 ± 2.0 °C for 5 days on a rotary shaker (150 rpm). Nitrogenase activity was assayed in the bacterial cultures using the acetylene reduction technique according to [46]. Nitrogenase activity was calculated based on the formula applied in [47].

#### 4.3.3. Production of IAA

Each bacterial isolate was grown in 20 mL of NB medium supplemented with 1 g of tryptophan per 1000 mL, and incubation on a rotary shaker was carried out at 30 ± 2 °C for 24 h [48]. Cultures were centrifuged at 10,000 rpm for 15 min at room temperature, and 2.0 mL of supernatant was added to 2.0 mL of Salkowski’s reagent. These mixtures were kept for 30 min in the dark, and the absorption was measured at 530 nm [49]. Concentrations of IAA (µg/mL) were calculated from the IAA standard curve.

#### 4.3.4. Detection of NH_3_

Ammonia (NH_3_) was detected according to the methods applied in [50]. Bacterial isolates were grown in 20.0 mL of peptone broth medium and incubated on a shaker at 30 ± 2 °C for 48 h. Incubated cultures were centrifuged at 10,000 rpm for 15 min. A total of 2 mL of the filtrate was mixed with 1.0 mL of Nessler’s reagent. The existence of varied light yellow to deep brown color indicated the successful synthesis of NH_3_.

### 4.4. Germination Test

The germination percentage of maize grains (Giza 310, triple hybrid white sweet corn, kindly obtained from the Agric. Res. Station, General Authority for Agrarian Reform, Ministry of Agriculture and Land Reclamation, Giza, Egypt) treated with the isolated N_2_-fixing bacteria was calculated. Maize grains were surface sterilized through soaking in 0.5% sodium hypochlorite solution for 3 min, rinsed 3 times with sterilized dH_2_O before planting. Sterilized maize grains were soaked in 7 d cultures for each isolate in a shaking incubator (150 rpm) at 30 ± 2 °C for 8 h. Grains were allowed to dry in the air before sowing. Grains immersed in sterilized dH_2_Owere acted as control. Ten grains were sown in small plastic bags filled with 80 g of sterilized clay soil. Each bag received 5.0 mL of the microbial inoculum as a booster dose. Trials were performed in triplicates for 10 days and watered when necessary. The germination percentage due to each isolate was calculated as follows:Germination percentage (%) = (No. germinated seeds/No. total seeds) × 100(2)

### 4.5. Identification of the N_2_-Fixing Isolates

Based on their optimization and germination test, one of the most efficient isolates was selected and subjected for identification via its cultural and biochemical traits. Identification was also confirmed based on the molecular test.

#### 4.5.1. Cultural and Biochemical Tests

According to their cultural and biochemical traits, the isolates was identified using Bergey’s Manual of Determinative Bacteriology. The sample was microscopically tested to describe the shape of the cells, endospore formation, Gram-staining reaction and motility. Growth in different Na Cl, temperature, and pH levels was also examined. Catalase activity was determined using 10% hydrogen peroxide (H_2_O_2_) solution for 24 h and recognized based on the production of air bubbles of oxygen gas [51]. Phosphate solubilization was determined through the spot inoculation of the bacterial isolate on plates containing Pikovskaya’s agar media, which were incubated at 30 ± 2 °C for 7 d. The appearance of a clear halo zone indicates P solubility [43]. For carbohydrate fermentation, slants contained glucose or fructose or mannitol broth media as the sole source of carbon. Bromothymol blue was inoculated in a Durham tubewith the bacterial isolate and incubated at 37 °C for 5–7 days. The appearance of a strong yellow color instead of blue in the slants and gas formation in the Durham tube indicated acid production and carbohydrate fermentation. If the broth media remained blue, non-fermentative culture was indicated.

#### 4.5.2. Molecular Test

To confirm the identification of the selected isolate, a 16S rRNA gene sequencing test was applied. Genomic DNA extraction of the isolates was carried out using Quick-DNA™ according to the protocol of the bacterial micro-prep kit procedure (Zymo-Spin™ IIC-XL). This technique was performed by Sigma, Cairo, Egypt. The rDNA gene was amplified through the Thermocycler using Maxima Hot Start PCR Master Mix (Thermo K1051, Fermentas, Waltham, MA, USA) according to the Applied Bio-Systems (ABI) 3730xl DNA sequences using primer (forward and reverse) pairs 1163F,5-GTGAGCGTCAGCTCGTAATA-3 and 1578R, 5-ACTAGCTAACAGGATGCACT-3 for *Azotobacter* sp. [52]. Sequencing was performed using PCR products at Biotech Company, Constance, Germany (GATC) according to the traditional Sanger technology with the new 454 technologies. The 16S rDNA sequence was compared to those of reference taxa obtained from sequences in the GenBank database using the National Institute of Technology and Evaluation Search Tool (NITE) (https://nite.go.jp/enLnbrc/, accessed on 20 September 2023). The phylogenetic analyses were conducted based on molecular evolutionary genetics analysis (MEGA6) [53]. For phylogenetic analysis, maximum parsimony and neighbor-joining trees were constructed from 1000 bootstrap replicates.

### 4.6. In Vivo Trials

To evaluate their capability, the studied N_2_-fixing strains, *A. salinestris* (the most efficient isolate) and *A. oryzea* SWERI 111 (kindly obtained from the bio-fertilizer production unit, Sakha Agriculture Research Station, Kafr El-Sheikh, Egypt), were inoculated (sole/dual) with the mineral N in pots and field trials. For expanding mass production, the isolated N_2_-fixing bacteria were cultured in NB media and incubated in a rotary shaker (125 rpm) at 30 ± 2 °C for 7 d. The total count of each bacterial strain was adjusted at 150 × 10^8^ CFU/mL and used to prepare their inoculum based on the methods described in [54]. Bacterial cultures were mixed and homogenized with sterilized peat at the rate of 2400 mL/1200 g peat/ha and 100 mL/50 g peat/pot. The peat–bacterial mixture was packed in polyethylene bags and kept at 4 °C in the refrigerator. Maize grains were thoroughly mixed with the peat-based bacterial inoculum after being wet with 10% sugar solution. Grain–peat–bacterial units were exposed to air, drying in the shade for 30 min before sowing [55].

#### 4.6.1. Pot Experiment

Under outdoor conditions, pots (30 cm in diameter) filled with 8 kg autoclaved clay soil (Table 7) during the 2020 season were used. Five peat-bacterial-maize grains for each pot were planted. Three replicates were applied for each treatment. Grains dipped in dH_2_O acted as the control. A total of 10 d later, each pot was inoculated with 20 mL of bacterial culture as a booster dose. Another 10 d later, pots received mineral N fertilizer in the form of urea (46.5% N). On the basis of the 100% mineral N equal 120 N-unit, 4 mineral N- concentrations of 100, 75, 50, and 25% represented by 4.34, 3.26, 2.17, and 1.08 g urea/pot, respectively, were used. The total N amount was divided into two halves, 10 days each. Irrigation and other practices were carried out as recommended. Some vegetative growth parameters, chlorophylls a-b, and total N content were determined 35 days after planting.

#### 4.6.2. Field Experiment

Under field conditions (Table 7), treatments were also tested during the 2021 season. Plots were made up of 5 ridges (5 m long and 60 cm wide for each ridge) planted in hills (30 cm apart) with three maize grains inoculated with the peat-based bacterial mixtures. Each treatment was represented by 3 plots. Grains dipped in dH_2_O acted as controls. A total of 15 d later, hills were reduced to one and two seedlings’ intervals and received peat–bacterial mixtures as a booster dose, raking the soil 5 cm next to the plants. For N mineral fertilization, tested doses were divided into two halves each, 15 d apart. Thus, plots received the first half of the mineral N 30 d after planting, while the second half was added 15 d after the first half. Thus, N levels of 100, 75, 50, and 25% represented by 619.34, 464.52, 309.67, and 154.84 kg urea ha^−1^ were, respectively, added as total amounts. Irrigation and other practices were also conducted as recommended for the maize production regime. Vegetative growth and yield parameters were measured 65 and 95 d after planting, respectively. The induction of polyphenol oxidase (PPO), peroxidase (PO), and proline was also evaluated under field conditions.

##### Plant Growth and Yield Parameters

In pots and field trials, vegetative growth parameters such as fresh and dry weight and leaf area were measured. Weighted fresh samples were dried at 60 °C for a time showing constant weight. Accordingly, dry matter of both leaves and shoots were calculated as follows:Dry matter (%) = (dry weight/fresh weight) × 100(3)

At harvest, the grain count per cob and 1000-kernel weight were determined.

##### Biochemical Measurements


*Chlorophyll*


Chlorophyll a and b were measured using a certain fresh flag leaf area of maize plants 35 and 65 days after planting in both pot and field trials, respectively. According to [56], chlorophyll contents were extracted through immersing the samples in 5 mL of N, N-dimethylformamide (dark conditions), which were measured at 647 and 664 nm using a spectrophotometer (Jenway 6105 UV-VIS, Bibby Scientific Ltd, Dunmow, Essex. UK). To calculate chlorophyll a and b per µg/mL, readings were used based on the following equations:Chl a = 12.46 × (A 664) − 2.49 (A 647) = µg/mL(4)
Chl b = −5.6 × (A 664) + 23.26 (A 647) = µg/mL(5)


*Proline*


To extract proline, 0.5 g of fresh leaf sample was immerged in 5.0 mL of 3% sulphosalicylic acid solution for 2 h. Then, 2.0 mL of the extract was mixed with 2.0 mL of glacial acetic acid and 2.0 mL of a mixture (1.25 g of ninhydrin, 30.0 mL of glacial acetic acid, and 20.0 mL of orthophospiric acid) for 1 h in a boiling water bath before being treated with 4.0 mL of toluene. Through taking the upper oily layer, absorbance was determined using a spectrophotometer at 520 nm [57]. Proline concentrations were calculated as µ mol g^−1^ according to the standard graph of proline estimation.


*Total nitrogen and protein*


Samples of maize grains were collected, dried in an oven at 70 °C, finely powdered, and then digested using the sulphuric-perchloric acid procedure based on the micro-Kjeldahl method as described in [58] as follows:N-content (% per g dry weight) = Total-N% × dry weight of plants/100(6)
Crude protein (% per g dry weight) = N-content% × 5.74(7)


*Enzymes activities*


The effects of the studied treatments on the activities of some oxidative enzymes of PPO and PO were assayed 65 d after planting in the leaves. To prepare crude enzyme extract, leaf samples were cleaned, and 0.5 g was fresh weighted and triturated in an ice-chilled mortar with liquid nitrogen gas, and the resulting powder was suspended in 5 mL of solution buffer (sodium phosphate, 0.1 M, pH 7.1). Samples were filtrated through cheesecloth and centrifuged at 12,000 rpm under cooling at 4 °C for 10 min. The supernatant fractions were kept frozen at −80 °C. Three replicates were processed in each case [59].

The enzyme activity of PPO was assayed according to the colorimetric procedures adopted for guayacol (C_6_ H_4_ (OH)_2_). Samples were maintained at −80 °C before the enzyme assay. For device calibration, a mixture consisting of 4.0 mL sodium phosphate buffer (0.1 M at pH 7.0), 1.0 mL guayacol, and 5.0 mL dH_2_O was prepared. For enzyme activity, 4.0 mL of the mixture was added and homogenized with 1.0 mL of the crude enzyme extract. Using a UV-VIS spectrophotometer T80 + (EPCRS, 016), absorbance was measured at an optical density (OD) of 495 nm, and data were recorded after 0, 30, 60, 90, 120, 150, and 180 s [37].

The enzyme activity of PO was assayed according to the colorimetric procedures adopted for Pyrogaloll, depending on the oxidation of the colorless Purpurogallin to exhibit brown coloration in the presence of H_2_O_2_. A combination consisting of 1.0 mL of (0.1 M) sodium phosphate-buffered solution, 1.0 mL of Pyrogaloll, and 1.0 mL of H_2_O_2_ was diluted to a 10.0 mL final volume using dH_2_O. Then, 0.2 mL of the crude enzyme extract was mixed with 3.0 mL of the combination. At 470 nm, the absorbance of the samples was measured, and the OD was recorded after 0, 30, 60, 90, 120, 150, and 180 s using a UV-VIS spectrophotometer T80+ (EPCRS, 016). The treatment contained all chemical reagents except the enzyme extract that served as a blank, which is used to reset or calibrate the device. The combined activity of both PPO and PO was calculated based on the formula applied in [60,61].

### 4.7. Statistical Analysis

Using the ‘COSTAT’ computer software program statistical tool, data were submitted for analysis of variance. Duncan’s multiple range tests were used to examine means, with a *p* value of 0.05 [62]. The data were presented as mean ± SE.

## 5. Conclusions

The utilization of N_2_-fixing bacteria was found to be an effective and environmentally safe alternative to chemical fertilizers, or at least reduce its use. The dual mixture of both bacteria, identified as A. salinestris OR512393 and A. oryzea, in the presence of 75% mineral N, was found to be the superior and promising treatment to enhance all growth and yield parameters of maize plants. Such treatment played a key role in improving soil fertility through increasing the availability of nutrients required for plant growth and minimizing the cost of maize productivity.

## Figures and Tables

**Figure 1 plants-12-03830-f001:**
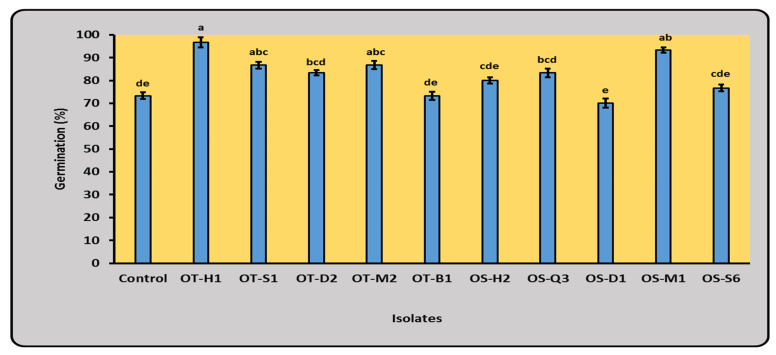
Effect of N_2_-fixing bacterial isolates on the germination percentage of maize grains in plastic bags. Data are presented as the mean ± SE; a–e: Duncan’s letters.

**Figure 2 plants-12-03830-f002:**
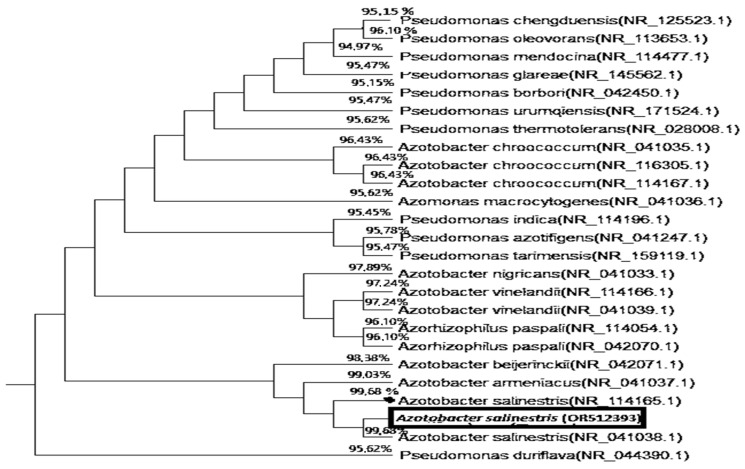
Polygenetic dendrogram resulting from distance matrix analysis of 16S rRNA sequences showing the position and the highest similarity of (OT-H1) *Azotoacter salinestris* strain OR512393 among the phylogenetic neighbors.

**Figure 3 plants-12-03830-f003:**
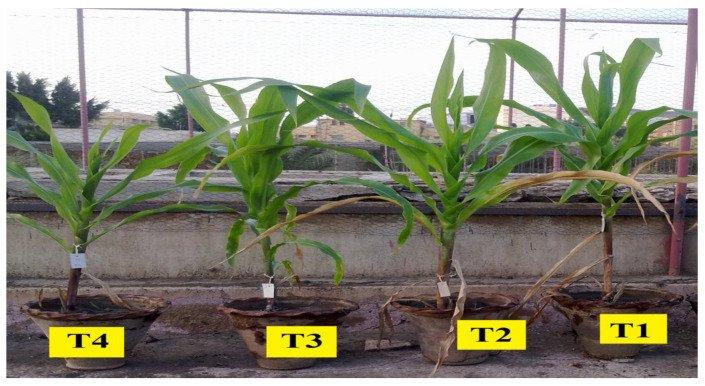
General photos for different doses of nitrogen and inoculation treatments on vegetative growth of maize plants after 60 days. T1–T13: as shown in Table 3.

**Figure 4 plants-12-03830-f004:**
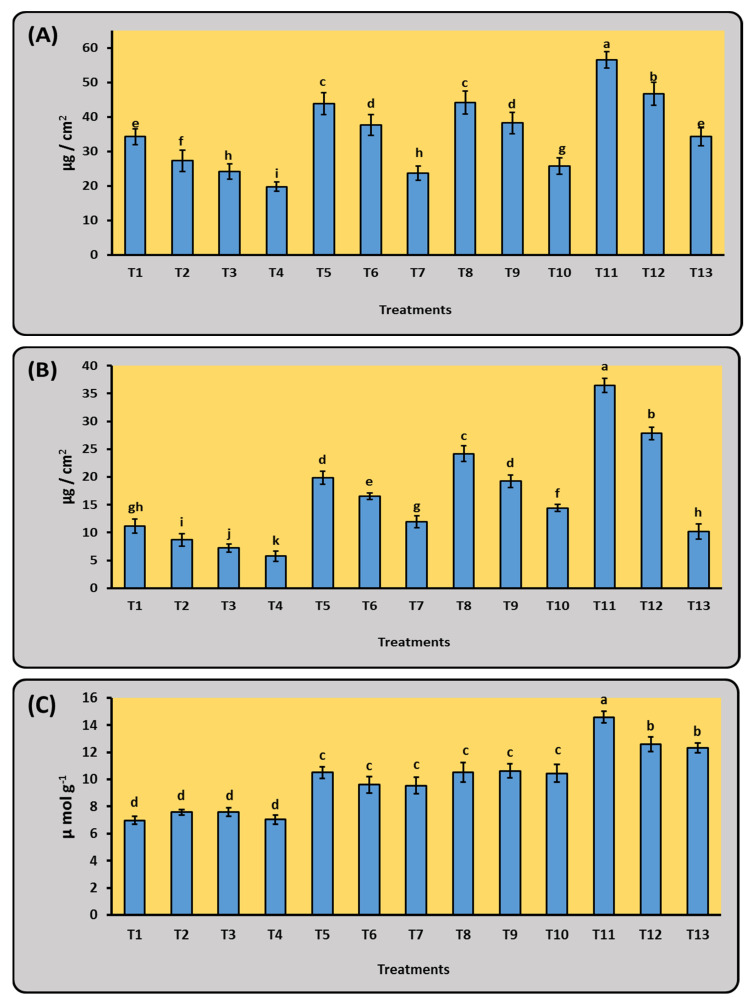
Effect of N_2_-fixing bacteria in the presence of different doses of mineral N on biochemical measurements (**A**) cholorophyll a, (**B**) cholorophyll b, and (**C**) proline of maize plants 65 days from planting under natural open field conditions. Data are presented as mean ± SE; a–k: Duncan’s letters; T1–T13: as shown in Table 3.

**Figure 5 plants-12-03830-f005:**
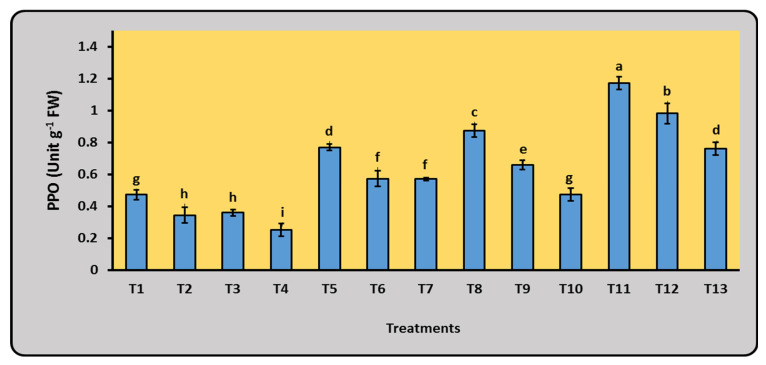
Enzyme activities of polyphenol oxidase (PPO) and peroxidase (PO) of maize flag leaf 65 days after planting under different doses of mineral N and bacterial inoculation. Data are presented as mean ± SE; a–i: Duncan’s letters; T1–T13: as shown in Table 3.

**Table 1 plants-12-03830-t001:** Performance of the tested bacterial isolates for total nitrogen content, nitrogen activity, production of indole acetic acid (IAA), and detection of NH_3_ in their cultures.

Isolate Code	Total Nitrogen Content (%)	Nitrogenase Activity (Nano-Mole mL^−1^ h^−1^)	IAA Production (µg mL^−1^)	NH_3_ Detection (+/-)
OT-H1	14.01 ± 1.23 ^a^	517 ± 22.19 ^a^	539 ± 32.89 ^a^	+
OT-S1	7.70 ± 1.09 ^d^	320 ± 17.93 ^e^	361 ± 26.19 ^e^	+
OT-D2	6.30 ± 0.97 ^f^	401 ± 24.28 ^c^	394 ± 25.10 ^c^	+
OT-M2	7.05 ± 1.11 ^e^	305 ± 19.20 ^f^	357 ± 27.02 ^e^	-
OT-B1	10.50 ±1.09 ^c^	378 ± 20.11 ^d^	303 ± 22.76 ^f^	-
OS-H2	4.90 ± 0.92 ^g^	242 ± 15.17 ^g^	263 ± 23.35 ^h^	-
OS-Q3	3.50 ± 0.69 ^i^	173 ± 11.30 ^i^	304 ± 23.11 ^f^	+
OS-D1	4.20 ± 0.93 ^h^	200 ± 17.90 ^h^	285 ± 20.87 ^g^	-
OS-M1	11.20 ± 1.12 ^b^	412 ± 22.19 ^b^	450 ± 28.29 ^b^	+
OS-S6	3.09 ±0.91 ^j^	201 ± 19.97 ^h^	379 ± 26.95 ^d^	-

+: Positive; -: negative; data are presented as the mean ± SE, ^a–i^: Duncan’s letters.

**Table 2 plants-12-03830-t002:** Cultural and biochemical characteristics of the most effective N_2_-fixing bacterial isolates.

Characteristics		OT-H1 Isolate
Colony		Watery and mucilaginous
Pigmentation transparent		Brown
Cell shape		Short rods to oval
Motility		+
Endospore formation		-
Gram reaction		-
Catalase induction		+
Phosphate solubilization		+
Utilization of carbon sources	Glucose	+
	Fructose	+
	Mannitol	+
pH	6.5	+
	7.5	+
	8.5	+
	9.5	+
Temperature °C	25	+
	30	+
	35	+
	40	+
NaCl %	2.0	+
	3.0	+
	4.0	+

**Table 3 plants-12-03830-t003:** Effect of N_2_-fixing bacteria in the presence of different doses of mineral N on dry matter of shoot and leaf, flag leaf area, chlorophylls a and b, and leaf total-N of maize seedlings 35 days from planting under pot conditions.

Treatment	Dry Matter of Shoot (%)	Dry Matter of Leaf (%)	Flag Leaf Area (cm^2^)	Chlorophyll (µg/cm^2^)		Total –N of Leaf (%)
				a	b	
**T1**	93.89 ± 6.12 ^ab^	55.12 ± 4.08 ^e^	121.86 ± 3.65 ^c^	20.21 ± 1.07 ^f^	5.57 ± 1.11 ^i^	0.84 ± 0.22 ^cd^
**T2**	81.44 ± 5.22 ^def^	49.16 ± 3.62 ^f^	68.31 ± 3.41 ^g^	14.95 ± 1.09 ^h^	8.5 ± 1.23 ^f^	0.63 ± 0.19 ^e^
**T3**	84.48 ± 4.92 ^cde^	43.20 ± 3.11 ^g^	43.39 ± 3.87 ^i^	12.84 ± 0.95 ^i^	6.39 ± 0.93 ^h^	0.42 ± 0.09 ^f^
**T4**	76.18 ± 3.88 ^f^	40.94 ± 3.01 ^g^	15.61 ± 1.97 ^k^	10.77 ± 0.89 ^j^	3.53 ± 1.01 ^k^	0.28 ± 0.11 ^g^
**T5**	91.06 ± 5.72 ^abc^	83.63 ± 4.12 ^b^	85.23 ± 4.12 ^f^	33.62 ± 1.02 ^c^	6.77 ± 1.02 ^h^	0.89 ± 0.19 ^bc^
**T6**	86.48 ± 4.19 ^b–e^	75.28 ± 4.85 ^c^	70.75 ± 3.82 ^h^	21.13 ± 1.01 ^e^	10.52 ± 1.47 ^d^	0.63 ± 0.18 ^e^
**T7**	80.62 ± 4.28 ^ef^	63.01 ± 3.86 ^d^	18.97 ± 1.11 ^jk^	17.93 ± 1.01 ^g^	8.03 ± 1.22 ^g^	0.42 ± 0.22 ^f^
**T8**	93.29 ± 5.38 ^ab^	84.70 ± 4.29 ^b^	84.01 ± 3.90 ^e^	35.19 ± 1.15 ^b^	5.84 ± 1.09 ^i^	0.98 ± 0.31 ^ab^
**T9**	92.70 ± 4.92 ^ab^	83.32 ± 4.91 ^b^	56.70 ± 3.86 ^h^	21.78 ± 0.85 ^de^	11.74 ± 0.87 ^c^	0.77 ± 0.21 ^d^
**T10**	88.77 ± 5.35 ^a–d^	86.26 ± 4.87 ^b^	21.90 ± 1.17 ^j^	18.01 ± 0.92 ^g^	9.47 ± 1.01 ^e^	0.56 ± 0.12 ^e^
**T11**	95.61 ± 6.09 ^a^	92.27 ± 4.95 ^a^	179.70 ± 6.97 ^a^	38.12 ± 1.09 ^a^	20.62 ± 1.65 ^a^	1.07 ± 0.31 ^a^
**T12**	95.04 ± 6.18 ^a^	83.62 ± 4.54 ^b^	147.42 ± 7.22 ^b^	33.13 ± 1.15 ^c^	13.42 ± 0.59 ^b^	0.84 ± 0.13 ^cd^
**T13**	92.25 ± 5.86 ^ab^	82.36 ± 4.81 ^b^	107.63 ± 5.96 ^d^	22.60 ± 0.98 ^d^	7.74 ± 0.55 ^g^	0.63 ± 0.19 ^e^

Values within the same vertical areas with the same letter are insignificantly different at 0.05 probability level according to Duncan’s multiple range test. Mix: dual mix of both bacterial strains. Data are presented as the mean ± SE. LSD: least significant difference; T1: 100% N; T2: 75% N; T3: 50% N; T4: 25% N; T5: 75% N + *A. salinestris*; T6: 50% N + *A. salinestris;* T7: 25% N + *A. salinestris*; T8: 75% N + *A. oryzea;* T9: 50% N + *A. oryzea;* T10: 25% N + *A. oryzea;* T11: 75% N + Mix; T12: 50% N + Mix; and T13: 25% N + Mix.

**Table 4 plants-12-03830-t004:** Effect of N_2_-fixing bacteria in the presence of different doses of mineral N on dry matter of shoot and leaf, flag leaf area, and leaf total-N of maize plants 65 days from planting under natural open field conditions.

Treatment	Plant Height (cm)	Dry Matter of Leaf (%)	Flag Leaf Area (cm^2^)	Total –N of Leaf (%)
**T1**	120.70 ± 5.66 ^d^	53.47 ± 2.34 ^f^	462.21 ± 23.11 ^g^	1.12 ± 0.11 ^cd^
**T2**	107.70 ± 5.18 ^f^	49.44 ± 2.87 ^g^	351.39 ± 26.19 ^j^	0.91 ± 0.19 ^e^
**T3**	97.70 ± 5.09 ^h^	45.64 ± 2.51 ^h^	301.86 ± 31.09 ^k^	0.70 ± 0.15 ^f^
**T4**	83.70 ± 4.87 ^k^	44.39 ± 1.97 ^h^	264.00 ± 34.29 ^l^	0.56 ± 0.13 ^g^
**T5**	123.70 ± 4.98 ^c^	75.10 ± 2.39 ^bc^	480.96 ± 35.98 ^f^	1.16 ± 0.23 ^bc^
**T6**	100.70 ± 4.93 ^g^	68.23 ± 2.65 ^d^	406.80 ± 39.10 ^i^	0.91 ± 0.27 ^e^
**T7**	86.70 ± 4.57 ^j^	58.97 ± 2.34 ^e^	353.73 ± 41.90 ^j^	0.72 ± 0.12 ^f^
**T8**	126.70 ± 6.09 ^b^	76.25 ± 2.86 ^bc^	573.87 ± 48.12 ^d^	1.26 ± 0.31 ^ab^
**T9**	106.70 ± 5.61 ^f^	74.42 ± 2.89 ^c^	495.60 ± 43.10 ^e^	1.05 ± 0.33 ^d^
**T10**	89.70 ± 5.18 ^i^	75.58 ± 2.61 ^bc^	436.77 ± 38.98 ^h^	0.84 ± 0.26 ^e^
**T11**	146.70 ± 6.09 ^a^	85.69 ± 2.75 ^a^	909.72 ± 66.76 ^a^	1.35 ± 0.34 ^a^
**T12**	126.70 ± 5.72 ^b^	77.78 ± 2.78 ^b^	746.40 ± 45.19 ^b^	1.12 ± 0.38 ^cd^
**T13**	115.70 ± 5.90 ^e^	76.20 ± 2.81 ^bc^	677.82 ± 47.71 ^c^	0.91 ± 0.27 ^e^

Values within the same vertical areas with the same letter are insignificantly different at 0.05 probability level according to Duncan’s multiple range test. Mix: dual mix of both bacterial strains. Data are presented as the mean ± SE. LSD: least significant difference. T1–T13: as shown in Table 3.

**Table 5 plants-12-03830-t005:** Grain number per cob, weight of 1000 kernels, N content, and crude protein of maize grains as affected by different levels of mineral N and N_2_-fixing bacteria 95 days after planting during the 2021 season under field conditions.

Treatment	Yield Parameters			
	Grain Count (cob^−1^)	1000 Kernel (g plot^−1^)	N Content	Crude Protein
			(% g^−1^ Dry Weight)	
**T1**	563.66 ± 41.19 ^b^	604.33 ± 44.10 ^c^	5.34 ± 1.02 ^i^	5.61 ± 1.01 ^d^
**T2**	402.00 ± 38.90 ^g^	473.00 ± 41.98 ^j^	3.29 ± 1.17 ^k^	2.70 ± 0.59 ^g^
**T3**	337. 30 ± 36.11 ^j^	450.00 ± 40.12 ^k^	2.82 ± 1.29 ^l^	2.20 ± 0.29 ^h^
**T4**	283.00 ± 28.98 ^m^	294.03 ± 29.90 ^m^	1.61 ± 1.02 ^m^	0.82 ± 0.19 ^i^
**T5**	475.33 ± 39.10 ^e^	594.00 ± 43.19 ^e^	6.80 ± 1.04 ^d^	7.03 ± 1.09 ^c^
**T6**	357.00 ± 35.39 ^h^	532.00 ± 44.67 ^g^	6.06 ± 1.28 ^g^	5.61 ± 0.99 ^d^
**T7**	312.00 ± 33.34 ^k^	486.00 ± 41.69 ^h^	4.03 ± 0.95 ^j^	3.40 ± 0.81 ^f^
**T8**	466.00 ± 40.19 ^f^	582.00 ± 47.89 ^f^	6.96 ± 1.02 ^c^	7.05 ± 1.11 ^c^
**T9**	346.66 ± 32.45 ^i^	422.00 ± 39.20 ^l^	6.52 ± 1.10 ^f^	4.79 ± 1.01 ^e^
**T10**	296.66 ± 27.45 ^l^	480.00 ± 38.98 ^i^	5.73 ± 0.89 ^h^	4.78 ± 0.92 ^e^
**T11**	694.00 ± 45.23 ^a^	754.33 ± 49.87 ^a^	7.73 ± 1.09 ^a^	10.14 ± 1.87 ^a^
**T12**	522.00 ± 41.98 ^c^	694.00 ± 50.10 ^b^	7.15 ± 1.18 ^b^	8.63 ± 1.29 ^b^
**T13**	485.00 ± 41.45 ^d^	601.00 ± 49.23 ^d^	6.68 ± 1.21 ^e^	6.98 ± 1.05 ^c^

Values within the same vertical areas with the same letter are insignificantly different at 0.05 probability level according to Duncan’s multiple range test. Mix: dual mix of both bacterial strains. Data are presented as the mean ± SE. LSD: least significant difference. T1–T13: as shown in Table 3.

**Table 6 plants-12-03830-t006:** The occurrence of N_2_-fixing bacterial isolates from various locations at Kafr El-Sheikh Governorate.

No	Isolates Code	Location	pH	EC (dS m^−1^)
**1**	OT-H1	Elhamoul (31°20′58″ N latitude and 31°09′01″ E longitude)	9.00 ± 1.22	2.70 ± 0.25
**2**	OT-S1	Sakha (31°04′26″ N latitude and 30°54′52″ E longitude)	8.67 ± 1.02	0.26 ± 0.12
**3**	OT-D2	Disouq (31°09′12″ N latitude and 30°39′45″ E longitude)	8.75 ± 1.23	0.83 ± 0.19
**4**	OT-M2	Motobas (31°16′44″ N latitude and 30°30′58″ E longitude)	8.10 ± 1.17	0.46 ± 0.11
**5**	OT-B1	Baltim (31°30′59″ N latitude and 31°06′07″ E longitude)	9.11 ± 1.31	1.03 ± 0.29
**6**	OS-H2	Elhamoul (31°20′58″ N latitude and 31°09′01″ E longitude)	9.00 ± 1.29	2.70 ± 0.25
**7**	OS-Q3	Qulin (31°04′34″ N latitude and 30°51′47″ E longitude)	8.16 ± 1.11	1.74 ± 0.31
**8**	OS-D1	Disouq (31°09′12″ N latitude and 30°39′45″ E longitude)	8.75 ± 1.09	0.83 ± 0.18
**9**	OS-M1	Motobas (31°16′44″ N latitude and 30°30′58″ E longitude)	8.10 ± 1.02	0.46 ± 0.14
**10**	OS-S6	Sidi Salim (31°07′52″ N latitude and 30°57′29″ E longitude)	8.59 ± 1.12	0.70 ± 0.19

Data are presented as the mean ± SE for 3 replicates.

**Table 7 plants-12-03830-t007:** Some physical and chemical analysis of soil used in pot and field experiments.

Mechanical Analysis (%)				Texture	pH (1:2.5)	EC (dSm^−1^)	OM (g Kg^−1^)	Available Elements (mg Kg^−1^)		
	Sand	Silt	Clay					N	P	K
**Pots**	7.4	7.5	85.1	Clayey	8.76	3.80	15.18	8.44	8.08	316.55
**Field**	3.1	7.7	89.2	Clayey	9.10	4.18	17.87	8.98	9.86	401.83

## Data Availability

Data are contained within the article.

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
