# Peer review of "Potential Impacts of Certain N2-Fixing Bacterial Strains and Mineral N Doses for Enhancing the Growth and Productivity of Maize Plants"

_plants, 2023, doi:10.3390/plants12223830_

Round 1
Reviewer 1 Report
Comments and Suggestions for Authors
The assesses the use of N2-fixing bacteria for enhancing the growth of maize plants. The work is novel and fits well within the scope of the study, and I believe it should be accepted as it is due to its high quality and scholarly merit.
Reviewer 2 Report
Comments and Suggestions for Authors
Abstract
1. The abstract is informative but could benefit from a more structured presentation for clarity.
2. Consider revising the sentences to improve readability and flow.
3. Specify the statistical analysis method used in the field trials for a more comprehensive understanding of the results.
4. Mention the sample size and replication details in both pot and field trials to establish the robustness of the findings.
5. Elaborate on how the N2-fixing bacteria affect maize growth and yield parameters. This could provide a deeper understanding of the study's significance.
6. The abstract mentions "genomic sequences of 16S rRNA," but clarifying how these sequences were used in the study might be beneficial.
7. Consider rephrasing or providing additional context for "tiny magnitude" regarding the differences observed with A. oryzea.
8. Include information about the duration of the field trials and pot experiments.
9. Proofread for typographical errors and grammar issues to enhance the overall professionalism of the abstract.
Introduction
1. The introduction is informative, but it could be improved by breaking it into smaller paragraphs for better readability.
2. In line 43, consider specifying which negative effects of chemical fertilizers the study aims to counteract.
3. In lines 46-47, provide more context on what "plant-friendly microorganisms" are and why they are important.
4. Clarify in line 50 what is meant by "actual agricultural situations" and how these microorganisms will be applied.
5. In lines 53-58, explain the significance of combining inoculations in more detail, especially regarding how it leads to better nutrient transformation and a stable environment.
6. In lines 60-62, specify the practical applications and benefits of Azospirillum and Azotobacter species in crop cultivation.
7. In line 64, provide more specific information on how much biological nitrogen fixation can be increased in maize, rice, and wheat when treated with Azospirillum and/or Azotobacter spp.
8. The last sentence in the introduction is somewhat unclear and could benefit from specifying what the study aims to achieve or what the specific research objectives are.
Results
1. Consider breaking down the text into smaller paragraphs for improved readability, especially after line 75.
2. Clarify the method used for the isolation of the N2-fixing bacteria and provide more context on how the sampling was conducted from the different maize fields.
3. Specify the units of measurement for the visualization of cultures on Ashby mannitol medium, and explain why a round brown halo of bubbles of nitrogen gas indicates a probability of being Azotobacter.
4. The section heading "2.1. Optimization of the N2-fixing isolates" could be more descriptive to give a clearer idea of what this section will cover.
5. In line 82, consider revising "high total N-content reached 10.50%" for clarity, specifying what "10.50%" represents.
6. In lines 88-93, provide a more detailed explanation of the germination test needs, the significance of germination percentage, and how it relates to the study's objectives. Also add the picture of this test because, as I understand, it plays a major role in selection.
7. In line 98, clarify the purpose of the identification process for the N2-fixing isolate.
8. Provide context or background information on why these specific tests were conducted and how the results will contribute to the study's objectives in lines 100-141.
9. In lines 135-141, explain the significance of the molecular test and its relation to the study's goals. Provide context on why Azotobacter salinestris strain OR512393 was of interest. I have a major issue based on 16S rRNA you can not identify any bacteria up to the species level. Kindly revise the results or use more housekeeping genes for the identification.
10. Consider breaking down the text into smaller paragraphs for improved readability after line 141.
11. In line 169, provide a brief summary or transition sentence to introduce the "pots trial" section.
12. Specify the units of measurement used in Table 3 for the various parameters and variables.
13. In lines 195-202, clarify the significance of the biochemical measurements, such as chlorophyll and proline content, and how these measurements relate to the study's objectives.
14. In lines 217-224, provide more context about the yield parameters, such as grain yield's significance and the study's implications. Kindly add the field image by a drone that can easily show the effect of the inoculum.
Discussion
1. Consider starting with an introductory sentence that explains the concept of the rhizosphere and its relevance to the study, which would provide a clearer context for the subsequent statements.
2. When describing the rhizosphere components (soil close to the roots, root surface, endorhizosphere), make sure to explain the significance of each component and its role in the study.
3. Elaborate on how root activities, specifically the exudation of organic substrates, influence microbiological activity in the rhizosphere. Explain the role of these organic substrates in more detail.
4. Specify the types of chemical elements and microbial interactions in the rhizosphere that are under consideration in this study.
5. To enhance the clarity of the text, consider breaking down complex sentences into smaller, more digestible parts.
6. In line 241, consider rephrasing for better flow, for example: "Hence, the current study aimed to isolate, screen, and select effective Azotobacter strains, assess their impact on plant growth, and understand the effects of combined A. oryzea inoculation on maize plant productivity."
7. Ensure consistency in the usage of "Azotobacter" or "Azotobacter species" throughout the text.
8. Instead of repeating the phrase "in the current study," consider rephrasing for variety, such as "In this investigation."
9. Provide more information regarding the specific organic substrates or substances involved in the production of plant growth-promoting substances, and how Azotobacter is related to these substances.
10. The last sentence of the paragraph could be more explicit about the objectives of the study. Consider rephrasing it to clearly state that the goal of the study is to assess the effects of Azotobacter on maize plant growth and productivity.
11. Specify the significance of "plant growth-promoting substances" and consider explaining their relevance to the study in more detail.
12. Clarify what is meant by "most N2-fixing bacteria" and specify the types of organic compounds produced by these bacteria.
13. In line 255, provide more context about the role of free-living microbes in soil and their relation to nitrogen fixation.
14. Clarify the nature and extent of the effects described as "moderate effects between the two poles."
15. Instead of "in absence," consider using "in the absence."
16. Elaborate on the preparation of "enriched samples" from saline soil and explain the relevance of this step to the study's objectives.
17. Explain how the results demonstrate that the role of N2-fixing bacteria enhances germination percentage and why this is significant.
18. Specify the nature of the "reminder isolates" and how their effects compare to the superior isolates.
19. Provide more context about the role of plant growth hormones, the antioxidant system, siderophores, and improved plant nutrition in plant growth.
20. Specify how Azotobacter helps plants minimize the negative effects of abiotic stresses and the mechanism involved in this process.
21. Clarify how the identification results are in full agreement with the mentioned studies, and provide more context on the role of cultural and biochemical properties in identification.
22. Clarify the significance of Azotobacter being Na+-dependent, N2-fixing, and a plant growth promoter, and how these traits are relevant to the study.
23. Explain how these results relate to the study's objectives and the characteristics of Azotobacter mentioned earlier.
24. Specify the characteristics of the isolated bacteria that relate to the plant growth-promoting traits, such as salt tolerance, N2-fixation, P-solubilization, and IAA production. Explain how these traits are relevant to the study and its objectives.
25. Provide more information about how the treatment's superiority relates to "enhancement of cell elongation or cell division."
26. Elaborate on the role of N-stimulating amino acids in forming proteins, activating enzyme systems, and their significance for cell division and elongation.
27. Provide more context for "microbe's capacity" and explain how it relates to microbial growth regulators and their effects on plant development.
28. Specify how the half-dose of chemical fertilizers in combination with biofertilizers impacts total nitrogen and chlorophyll pigments, and explain the significance of these findings.
29. Clarify what "efficiency of the superior treatment" refers to and how it is explained.
30. Explain the concept of "microbe's capacity" in more detail and how it relates to photosynthesis, nutrient translocation, and nutrient accumulation.
31. Clarify the role of PGPR in enhancing stomata conductance and chlorophyll content index in maize seedlings, providing more context for the reader.
32. Specify the relationship between diastrophic bacteria, leaf N-levels, and the use of chemical fertilizers.
33. Clarify what "diastrophic inoculants" refer to and their relevance to sustainable agriculture.
34. Explain the mechanisms associated with enhancing plant development and increasing dry matter, water absorption, and yield related to Azospirillum.
35. In line 285, clarify how Azospirillum influences biomass production and enzyme activities, and why this is significant.
36. Consider providing a transition sentence at the end of this section to connect it with the next section.
37. Specify the significance of high proline levels in maize plants and explain how it is related to environmental changes.
38. Elaborate on how enzyme activities and proline content contribute to helping plants maintain water homeostasis, maintain turgor pressure, and resist osmotic stress.
39. Explain the role of auxin-producing PGPR in enhancing the growth
MM
4.1. Soil sampling:
- In the first sentence, it would be clearer to mention that the soil samples were collected from the rhizosphere of maize cultivations.
- It's better to provide the source of the methods used to estimate pH and EC (Reference [42]).
4.3. Optimization of the N2-fixing isolates:
- Clarify the goals of this optimization process.
- Explain why these specific characteristics (Total nitrogen content, Nitrogenase activity, IAA production, NH3 detection) are relevant and valuable.
4.3.1. Total nitrogen content:
- Provide more context about why measuring total nitrogen is important.
- Use bullet points for the formula explanation to improve readability.
4.3.2. Nitrogenase activity:
- Clarify why Nitrogenase activity is significant.
4.3.4. Detection of NH3:
- Clarify what is meant by "varied light yellow to dark brown color" - what specific ranges or values are considered positive?
4.4. Germination test:
- Provide context on why germination tests are conducted.
- Explain the significance of using maize grains for this test and the variety name.
4.5. Identification of the N2-fixing isolates:
- Clearly state the criteria used for the selection of the most efficient isolate.
- Revised "Cultural and biochemical tests" and "Molecular test" more clearly and concisely.
4.5.2. Molecular test:
- Clarify why molecular tests are essential for identification, and why only one gene is used.
- Provide more details about the molecular methods and the specific primer sequences.
4.6.1. Pot experiment:
- Explain the purpose of the pot experiments and how they relate to the overall study.
- Break down the steps in bullet points for clarity.
4.6.2. Field experiment:
- Provide context about the field experiments.
- Describe the plot design and size. All the information and images as results.
- Organize the subsections "Plant growth and yield parameters" and "Biochemical measurements" more clearly.
4.6.2.1. Plant growth and yield parameters:
- Clarify what specific growth and yield parameters are measured.
- Use bullet points or a structured format to present the data collection process.
4.6.2.2. Biochemical measurements:
- Provide more information on why these specific enzymes and measures are relevant.
- Organize the enzyme activities more clearly, possibly using bullet points.
4.7. Statistical analysis:
- Mention the statistical tool used and how it was applied.
- Consider using a bullet point format to describe the statistical analysis.
5. Conclusions:
- Summarize the key findings and their implications.
- Provide a clear take-away message for the reader.
Figure 2: Increase the space between bacterial sequence and add the scale. Add space after bacterial name.
Table 1. What is you added after mean±??(SD or SE), which Statistical analysis you performed.
Table 3: Add SD with mean, also add the image of pot experiment.
Table 4: Add SD with mean.
Table 5: Add SD with mean, also add the image of field experiment.
Table 6: Add SD with mean for pH and EC, or how many number used for these analysis.
Table 7: It is hard to accept any a data without replicate, due to lack of replication this manuscript may be rejected.
Comments on the Quality of English LanguageProofreading needed.
Reviewer 3 Report
Comments and Suggestions for Authors
Title :- Potential impacts of certain N2-fixing bacterial isolates and mineral-N doses for enhancing growth and productivity of maize plants
General comments :- well written, but presentation is not good. Need reshaping
Abstract :- more than 300 words abstract is not justified. Reduce the length, just add significant results. Several abbreviations given, some explained some not. Should be explained for the readers.
Keywords:- OK
Introduction:- very short, not justified in the context, the authors can not judge that all the readers are specialized in the subject. It should be around 1000 words, covering all the aspects giving the importance of N2 fixing bacteria, then significance of specific isolates, and mineral dozes, then significance of maize, as already given, but the context should be worldwide not country centric.
There is no hypothesis, no client objectives, no prospective,
Overall very poor introduction.
Results:- Ok but too expected and not novel
Discussion :- short and significant but fine
Methods :- Old but standard OK
Conclusion :- redundant and donot add any significance
Comments on the Quality of English Languageok
Round 2
Reviewer 3 Report
Comments and Suggestions for Authors
well revised
Comments on the Quality of English Languageok
